# Boosting targeted genome editing using the hei-tag

**Thomas Thumberger[1†], Tinatini Tavhelidse-Suck[1,2†], Jose Arturo Gutierrez-Triana[1†‡], Alex Cornean[1,2], Rebekka Medert[2,3,4], Bettina Welz[1,2,4], Marc Freichel[3,4], Joachim Wittbrodt[1,4]***

[1]Centre for Organismal Studies (COS), Heidelberg University, Heidelberg, Germany; [2]Heidelberg Biosciences International Graduate School (HBIGS), Heidelberg, Germany; [3]Institute of Pharmacology, Heidelberg University, Heidelberg, Germany; [4]DZHK (German Centre for Cardiovascular Research), partner site Heidelberg/Mannheim, Heidelberg, Germany

**Abstract** Precise, targeted genome editing by CRISPR/Cas9 is key for basic research and translational approaches in model and non-model systems. While active in all species tested so far, editing efficiencies still leave room for improvement. The bacterial Cas9 needs to be efficiently shuttled into the nucleus as attempted by fusion with nuclear localization signals (NLSs). Additional peptide tags such as FLAG- or myc-tags are usually added for immediate detection or straightforward purification. Immediate activity is usually granted by administration of preassembled protein/RNA complexes. We present the 'hei-tag (high efficiency-tag)' which boosts the activity of CRISPR/Cas genome editing tools already when supplied as mRNA. The addition of the hei-tag, a myc-tag coupled to an optimized NLS via a flexible linker, to Cas9 or a C-to-T (cytosine-to-thymine) base editor dramatically enhances the respective targeting efficiency. This results in an increase in bi-allelic editing, yet reduction of allele variance, indicating an immediate activity even at early developmental stages. The hei-tag boost is active in model systems ranging from fish to mammals, including tissue culture applications. The simple addition of the hei-tag allows to instantly upgrade existing and potentially highly adapted systems as well as to establish novel highly efficient tools immediately applicable at the mRNA level.

**\*For correspondence:**
jochen.wittbrodt@cos.uni-heidelberg.de

[†]These authors contributed equally to this work

**Present address:** [‡]Escuela de Microbiología, Facultad de Salud, Universidad Industrial de Santander, Bucaramanga, Colombia

## Editor's evaluation

This manuscript describes the addition of a short tag on the Cas9 nuclease as a means to improve genome editing efficiency. Importantly, the authors have tested their approach on several genomic targets, model organisms, and Cas9 derivative engineering tools. Overall, these findings support the possible general applicability of this tag for improving the outcomes of a wide range of modern Cas9 based applications, including Base Editors. Adding to a recent report of improving Prime Editing by optimising the NLS, this paper reinforces the notion that there is still unexplored space in altering genome engineering activity in a modular way.

## Introduction

In the last decade, the CRISPR/Cas9 system and its derivatives facilitated and revolutionized genome editing across all phyla (**Nidhi et al., 2021**). The efficiency of editing crucially depends on the on-site activity of the particular Cas9 enzymes used (usually *Streptococcus pyogenes Cas9*, *SpCas9*) in the nucleus. State-of-the-art Cas9 variants differ by peptide tags added to the N- and C-termini of the respective endonuclease resulting in reported different activities (**Liu et al., 2021**; **Zhang et al.,**

**eLife digest** The genetic code stored within DNA provides cells with the instructions they need to carry out their role in the body. Any changes to these genes, or the DNA sequence around them, has the potential to completely alter how a cell behaves.

Scientists have developed various tools that allow them to experimentally modify the genome of cells or even entire living organisms. This includes the popular Cas9 enzyme which cuts DNA at specific sites, and base editors which can precisely change bits of genetic code without cutting DNA. While there are lots of Cas9 enzymes and base editors currently available, these often differ greatly in their activity depending on which cell type or organism they are applied to.

Finding a tool that can effectively modify the genome of an organism at the right time during development also poses a challenge. All the cells in an organism arise from a single fertilized cell. If this cell is genetically edited, all its subsequent daughter cells (which make up the entire organism) will contain the genetic modification. However, most genome editing tools only work efficiently later in development, resulting in an undesirable mosaic organism composed of both edited and non-edited cells.

Here, Thumberger et al. have developed a new 'high efficiency-tag' (also known as hei-tag for short) that can enhance the activity of gene editing tools and overcome this barrier. The tag improves the efficiency of gene editing by immediately shuttling a Cas9 enzyme to the nucleus, the cellular compartment that stores DNA. In all cases, gene editing tools with hei-tag worked better than those without in fish embryos and mouse cells grown in the laboratory. When Cas9 enzymes connected to a hei-tag were injected into the first fertilized cell of a fish embryo, this resulted in an even distribution of edited genes spread throughout the whole organism.

To understand how a gene affects an organism, researchers need to be able to edit it as early in development as possible. Attaching the 'hei-tag' to already available tools could help boost their activity and make them more efficient. It could also allow advances in medical research aimed at replacing faulty genes with fully functioning ones.

*2014*). Employed tags usually comprise diverse nuclear localization signals (NLSs) and epitope tags (e.g. FLAG, Myc, HA) for potential protein purification or visualization. To achieve nuclear localization of the Cas9 enzyme, the monopartite NLS originating from the SV40 large T-antigen (*Kalderon et al., 1984*) or a bipartite NLS discovered in *Xenopus* nucleoplasmin is routinely employed (*Dingwall et al., 1988*). However, the nuclear localization activity of commonly used NLSs is tightly controlled during early development (*Poon and Jans, 2005*) and is first detectable during gastrulation. In fish embryos, an optimized artificial NLS (*Inoue et al., 2016*) (oNLS) facilitates prominent nuclear localization already immediately after fertilization, while the SV40 NLS acts most prominently much later and facilitates nuclear localization approximately at the 1000-cell stage. For high targeting efficiency with low mosaicism, a peak activity should be achieved in the zygote or at early cleavage stages. Here, we present the hei-tag, a short bipartite tag composed of a myc-tag and optimized NLSs at the N- and C-termini, that boosts Cas9 or cytosine-to-thymine (C-to-T) base editor-mediated targeted genome editing *in organismo* and cell culture.

## Results

Assessing the genome editing efficiency requires a reliable and quantitative readout based on an apparent phenotype. We established a quantitative assay for loss-of-eye pigmentation to address the activity of different Cas9 variants in two teleost model systems, medaka (*Oryzias latipes*) and zebrafish (*Danio rerio*) covering a wide evolutionary distance of 200 million years (*Furutani-Seiki and Wittbrodt, 2004*). Our assay on retinal pigmentation provides a highly reproducible quantitative readout for the loss of the conserved transporter protein *oculocutaneous albinism type 2* (*oca2*), required for melanin biosynthesis (*Figure 1a*). Only its bi-allelic inactivation results in the loss of pigmentation of eyes and skin (*Lischik et al., 2019*). A prominent knock-out phenotype thus can either result from a single to few early events, or from many events at subsequent developmental stages. Although phenotypically indifferent, the allele variance (genetic mosaicism) reflects the time point of action.

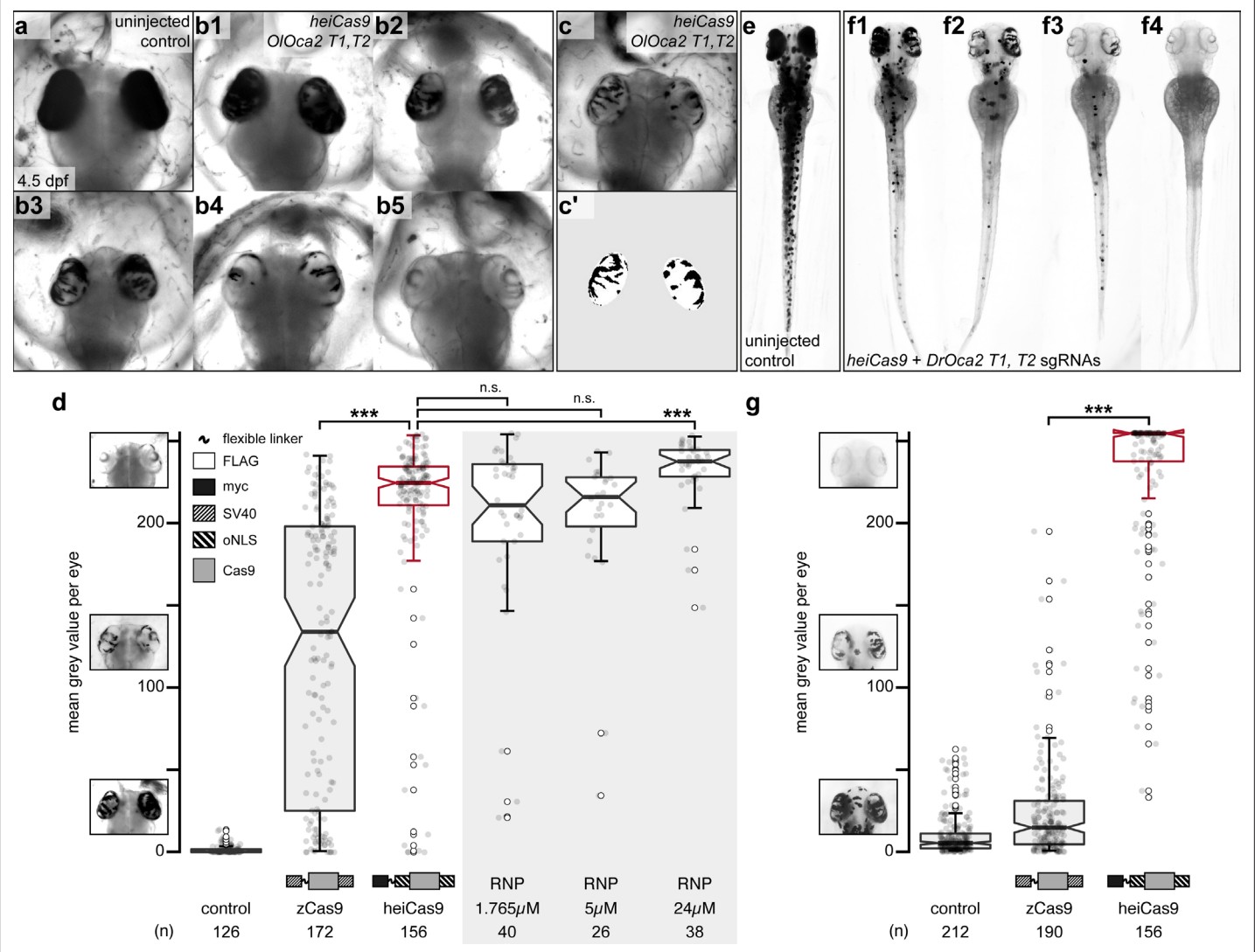

**Figure 1.** heiCas9 exhibits outstanding bi-allelic targeting activity in fish. Phenotypic range and quantification of *OlOca2 T1, T2,* and *DrOca2 T1, T2* sgRNAs/*Cas9 variant* and sgRNA/Cas9 protein complex (ribonucleoprotein [RNP])-mediated loss of pigmentation in medaka (**a–d**) and zebrafish (**e–g**) at high concentrations. (**a**) Fully pigmented eyes in uninjected control medaka embryo at 4.5 dpf. (**b1–b5**) Range of typically observed loss-of-pigmentation phenotypes upon injection with 150ng/µl *heiCas9* mRNA and 30ng/µl *OlOca2 T1, T2* sgRNAs. The observed phenotypes range from almost full pigmentation (**b1**) to completely unpigmented eyes (**b5**). (**c**) Minimum intensity projection of a medaka embryo at 4.5days after injection with 150ng/µl *heiCas9* and 30ng/µl *OlOca2 T1, T2* sgRNAs. (**c'**)Locally thresholded pigmentation on elliptical selection per eye (same embryo as in c). (**d**) Quantification of mean gray values (0 = fully pigmented, 255 = completely unpigmented) of individual eyes from Oca2 knock-out medaka crispants co-injected with 30ng/µl *OlOca2 T1, T2* sgRNAs and 150ng/µl mRNAs of *zCas9* and *heiCas9* (red) compared to RNP injections (concentrations indicated). Medians: uninjected control = 0.4; zCas9 = 134.5; heiCas9 = 225.3; 1.765µM RNP = 211.1; 5µM RNP = 216.2; 24µM RNP = 237.8. Note: highly significant pigment loss (70% increase) in heiCas9 vs. zCas9 crispants (p = 1.1e-25); heiCas9 reaches the same knock-out efficiency compared to RNP injections with only significant differences at highest RNP concentrations (24µM). (**e**) Fully pigmented uninjected control zebrafish embryo at 2.5 dpf. (**f1–f4**) Range of typically observed loss-of-pigmentation phenotypes upon injection with 150ng/µl *heiCas9* mRNA and 30ng/µl *DrOca2 T1, T2* sgRNAs. The observed phenotypes range from almost full pigmentation (**f1**) to completely unpigmented eyes and body (**f4**). (**g**) Quantification of mean gray values of individual eyes from o*ca2* knock-out zebrafish embryos co-injected with 30ng/µl *DrOca2 T1, T2* sgRNAs and 150ng/µl mRNAs of *zCas9* and *heiCas9* (red), respectively. Medians: uninjected control = 5.3; zCas9 = 14.7; heiCas9 = 254.6. Note the very highly significant pigment loss (17-fold increase) in heiCas9 vs. zCas9 crispants (p = 2.1e-56). dpf, days post fertilization; mean gray values ranged from 0, that is, fully pigmented eye to 255, that is, complete loss of pigmentation; n, number of eyes analyzed. Bold line, median. Statistical analysis performed in R, pairwise Wilcoxon rank sum test, Bonferroni corrected.

The online version of this article includes the following source data and figure supplement(s) for figure 1:

**Source data 1.** Raw data for quantifications shown in *Figure 1d and g*.

**Figure supplement 1.** Identification of the hei-tag.

*Figure 1 continued on next page*

*Figure 1 continued*

**Figure supplement 1—source data 1.** Raw data for quantifications shown in *Figure 1—figure supplement 1*.

**Figure supplement 2.** Survival and abnormality rate of Cas9 mRNA and ribonucleoprotein (RNP) injections.

State-of-the-art protocols employ high concentrations of Cas9 and respective sgRNAs to ensure efficient on-site editing. To facilitate uniform Cas9 action, we followed our successful mRNA injection protocol (*Gutierrez-Triana et al., 2018*). One-cell stage medaka embryos were co-injected with sgRNAs targeting the *oca2* gene (*OlOca2 T1, T2*) together with mRNA encoding a Cas9 endonuclease and mRNA encoding the green fluorescent protein (GFP) as injection tracer. Injected embryos were fixed at 4.5 days post fertilization (dpf) (*Iwamatsu, 2004*) well after the onset of pigmentation in control injections and subjected to image analysis (*Figure 1b*). In brief, the eyes were segmented, (residual) pigmentation was thresholded (*Figure 1c–c'*) and quantified according to mean gray values (0, i.e. fully pigmented, 255, i.e. completely unpigmented, *Figure 1d*).

We first established the base activity level for the assay at standard conditions with high molar excess (150 ng/µl concentration) and determined the activity of a Cas9 variant codon optimized for zebrafish, that is, a Cas9 carrying an SV40 NLS at the N- and C-terminus (nls-zCas9-nls, hereinafter: zCas9, Plasmid #47929 Addgene, *Jao et al., 2013*). The analysis of medaka *oca2* knock-out embryos injected with *zCas9* revealed bi-allelic inactivation events of the *oca2* gene, yet with a strong overall variability as apparent by patchy unpigmented domains in the eyes (median of mean gray values = 134.5 compared to uninjected controls, median = 0.4; *Figure 1d*). This patchy distribution of small, unpigmented areas indicated that bi-allelic targeting occurred only in few cells at later stages of development. To address whether different peptide domains (NLSs, Myc-tag, amino acid linkers) flanking the Cas9 enzyme enhance the targeting efficiency, we performed a permutation screen with Cas9 variants carrying these domains at different positions, which resulted in the identification of the 'hei-tag' (*Figure 1—figure supplement 1*). The hei-tag comprises a myc-tag connected via a flexible linker to an oNLS at the N-terminus complemented by a second oNLS fused to the C-terminus of a mammalian codon-optimized *Cas9* (see *Supplementary file 1* for sequence) and in this conformation displayed highest editing activity. Any alteration of those domains in relative order or sequence negatively impacted on editing efficiency compared to the hei-tag (*Supplementary file 2*).

When assessing the activity of the resulting heiCas9 at high molar excess (standard conditions, 150 ng/µl), heiCas9 displayed a 70% increase in bi-allelic targeting efficiency vs. the reference zCas9 (median zCas9 = 134.5, heiCas9 = 225.3; *Figure 1d*) in medaka. Embryos co-injected with *heiCas9* mRNA and sgRNAs against *oca2* essentially lost pigmentation. The observed absence of pigmentation argues for an early time point of action due to high activity and efficient nuclear translocation of the tagged heiCas9 variant already at the earliest cleavage stages. In developing organisms, the time point of genome editing essentially impacts on the allele variance, that is, the number of alleles established by the targeting attempt. To immediately provide a functional editing machinery, preassembled ribonucleoproteins (RNPs) containing Cas9 protein and guide RNA are popular, employing high molar excess/high concentrations of Cas9 (*Kroll et al., 2021*; *Wu et al., 2018*). Strikingly, the editing efficiency of injected *heiCas9* mRNA was fully comparable to such RNP approaches (*Figure 1d*, *Figure 1—figure supplement 2*).

To address whether the enhancement by hei-tag fusion to Cas9 is applicable to different models, we next compared the activities of the zCas9 and heiCas9 in a second, evolutionarily distant fish species *D. rerio* (zebrafish) targeting the orthologous *oca2* gene (sgRNAs *DrOca2 T1, T2*; *Hammouda et al., 2019*). Injected and control embryos were fixed well after the onset of pigmentation at 2.5 dpf (*Kimmel et al., 1995*; *Figure 1e–f*) and subjected to the quantitative assay for eye pigmentation described above. Taking the activity of zCas9 as base level (median = 14.7), heiCas9 delivered an outstanding targeting efficiency (median = 254.6), reflecting a 17-fold increase (p = 2.1e-56) (*Figure 1g*, *Figure 1—figure supplement 2*). Similar to the results in medaka, yet even more pronounced, nearly unpigmented embryos were obtained with the heiCas9, arguing for highly efficient, early targeting. Taken together, addition of the hei-tag to a mammalian codon-optimized Cas9 resulted in the highly efficient heiCas9, which boosted the targeting efficiency 17-fold, even when used at saturating concentrations. It prominently inactivated both alleles of the targeted *oca2* locus, with a putatively early onset of action upon injection of *heiCas9* mRNA and the respective sgRNAs at the one-cell stage.

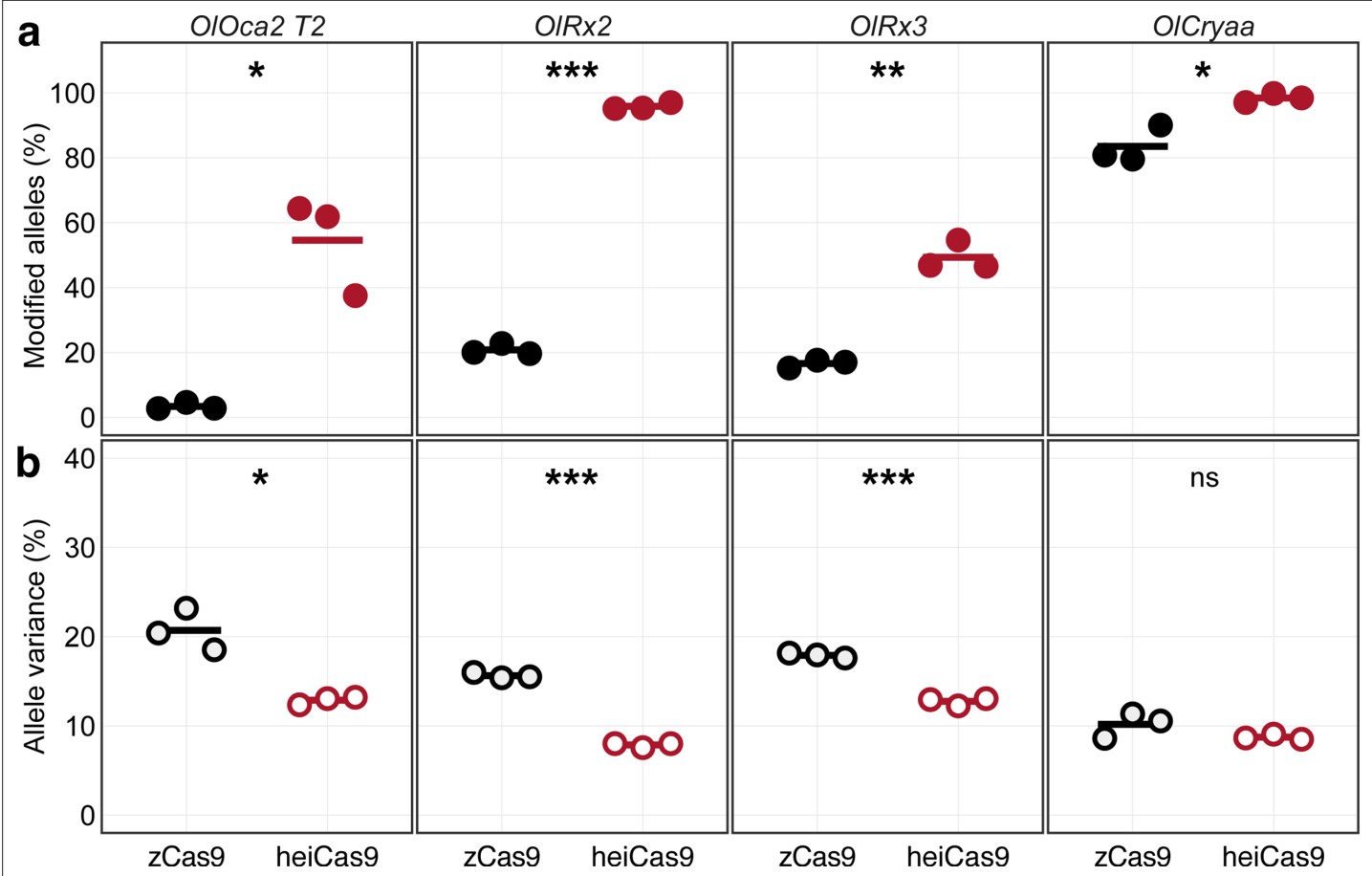

**Figure 2.** Increased knock-out activity and reduced allele variance in heiCas9 crispants. Multiplexed injections with 15ng/µl mRNA of *zCas9* or *heiCas9* (red) mRNA and 12.5ng/µl per sgRNA targeting exonic sequences in *oculocutaneous albinism type 2* (*oca2*; *OlOca2 T2*), the start codons of the *retina-specific homeobox transcription factor 2* (*rx2*; *OlRx2*) and of the *alpha a crystallin* (*cryaa*; *OlCryaa*), as well as an intronic sequence in *rx3* (*OlRx3*). Illumina sequencing performed on three biological replicates (eight embryos each) per targeted locus. (**a**) Increased knock-out efficiency in heiCas9 crispants as shown by proportion of modified over all Illumina sequencing reads per replicate and locus. (**b**) Reduced allele variance in heiCas9 crispants as shown by abundance of specific allele divided by all modified alleles per replicate and locus. Bold line, mean values of zCas9 (black) and heiCas9 (red). Total aligned Illumina reads analyzed: *OlOca2*: zCas9 = 194,931, heiCas9 = 180,222; *OlRx2*: zCas9 = 224,146, heiCas9 = 269,103; *OlRx3*: zCas9 = 195,248, heiCas9 = 175,044; *OlCryaa*: zCas9 = 209,573, heiCas9 = 200,448. Statistical analysis performed in R, Student's t-test.

The online version of this article includes the following source data and figure supplement(s) for figure 2:

**Figure supplement 1.** Mode of editing of all modified alleles.

**Figure supplement 1—source data 1.** Raw data for barplots shown in *Figure 2—figure supplement 1*.

To address whether the high targeting efficiency of heiCas9 was conveyed by the high molar excess employed or was possibly restricted to the *oca2* locus, we turned to a multiplexing regime at 10-fold reduced concentrations of the Cas9 variants employed. We targeted four different genomic loci with four different sgRNAs: exonic targeting of *oca2* (*OlOca2 T2*), targeting of the start codon of the *retina-specific transcription factor 2* (*rx2*; *Stemmer et al., 2015*), and the *crystallin alpha a* (*cryaa*; *Stemmer et al., 2015*) as well as intronic targeting of *rx3* (*Zilova et al., 2021*). Medaka one-cell stage embryos were co-injected with a mix of 12.5 ng/µl per sgRNA, the 10-fold reduced (15 ng/µl) *zCas9* or *heiCas9* mRNA and 20 ng/µl *mCherry* mRNA as injection tracer.

For each multiplexing experiment, the genomic DNA of three pools each containing eight randomly picked crispants was extracted at 4 dpf and subjected to allele-specific genotyping via Illumina sequencing. In the multiplexing approaches, a total of 823,898 reads for the zCas9 and 824,817 reads for the heiCas9, compared to 711,739 control reads, were analyzed (*Supplementary file 3*, *Figure 2—figure supplement 1*). In all cases, heiCas9 performed dramatically better than the reference zCas9 (*Figure 2a*; mean percentage of modified alleles zCas9 [black dots] vs. heiCas9 [red dots]:

*OlOca2*: 3.38% vs. 54.59%, p = 0.026; *OlRx2*: 20.82% vs. 95.85%, p = 3.2e-06; *OlRx3*: 16.61% vs. 49.36%, p = 0.0041; *OlCryaa*: 83.50% vs. 98.44%, p = 0.039). Strikingly, although the overall targeting efficiency was consistently higher as reflected by the high percentage of edited alleles (*Figure 2a*), at the same time the allele variance was reduced in all cases when using heiCas9 (*Figure 2b*; mean percentage of allele variance: zCas9 [black hollow dots] vs. heiCas9 [red hollow dots]: *OlOca2*: 20.71% vs. 12.87%, p = 0.025; *OlRx2*: 15.63% vs. 7.86%, p = 7.6e-06; *OlRx3*: 17.91% vs. 12.75%, p = 0.00021; *OlCryaa*: 10.17% vs. 8.74%, p = 0.22). This reduced allele variance for all multiplexed loci indicates an early editing by heiCas9. Given this and the overall higher targeting efficiency in all loci analyzed in the multiplexing approach, heiCas9 outperformed zCas9. It resulted in a massive performance boost, which was partially masked at saturating conditions, and now became fully apparent. The high efficiency of heiCas9 thus allows efficient editing at low concentrations with the potential to reduce off-target effects. Whether this putative reduction of off-targets is (over-)compensated by the efficient nuclear localization needs to be assessed by whole-genome sequencing approaches in the future.

While the early onset of action is required for uniform editing in developing organisms, cell culture approaches demand efficient translocation of the sgRNA/Cas9 complex in a large number of cells. To validate the range of action on the one hand and to address the relevance of the hei-tag in a mammalian setting, we expanded the scope of the analysis to mammalian cell culture. We focused on mRNA-based assays and compared the activity of heiCas9 to state-of-the-art Cas9 variants, that is, the commercially available *GeneArt CRISPR nuclease* as well as a mammalian codon-optimized Cas9 (*JDS246-Cas9,* Addgene #43861) in mouse SW10 cells. We assessed the respective genome editing efficiencies by independent and complementary tools, the Tracking of Indels by Decomposition (TIDE) analysis (*Brinkman et al., 2014*) as well as by Inference of CRISPR Editing (ICE) (*Hsiau et al., 2018*). Both approaches decompose the mixed Sanger reads of PCR products spanning the CRISPR target site and compute an efficiency score as well as the distribution of expected indels. To target the murine *Periaxin* (*Prx*) locus, mouse SW10 cells were co-transfected with *MmPrx* crRNA/

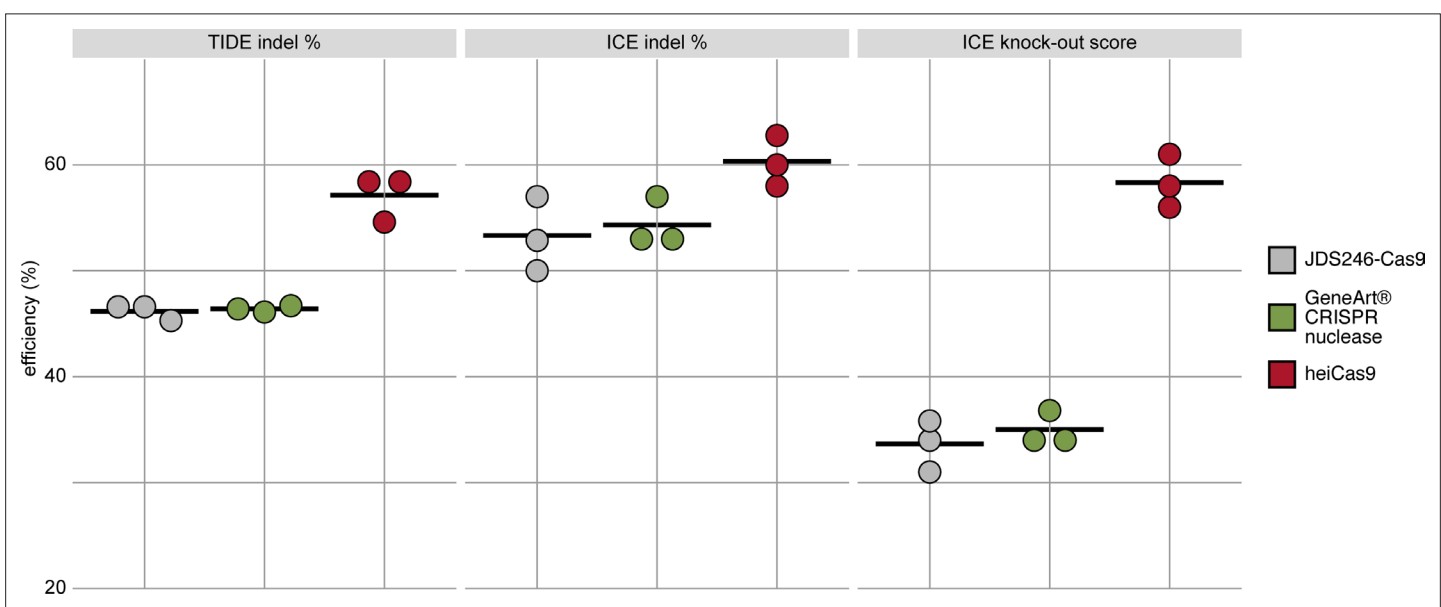

**Figure 3.** heiCas9 consistently exhibits high genome editing efficiency in mammalian cells. Mouse SW10 cells were co-transfected with *MmPrx* crRNA and mRNAs of *JDS246-Cas9, GeneArt CRISPR nuclease,* and *heiCas9,* respectively. Genome editing efficiency was assessed by Tracking of Indels by Decomposition (TIDE) and Inference of CRISPR Editing (ICE) tools. ICE knock-out score represents proportion of indels that indicate a frameshift or≥21bp deletion. Data points represent three biological replicates, black line indicates respective mean: TIDE indel %: JDS246-Cas9 = 46.2; GeneArt CRISPR nuclease = 46.4, heiCas9 = 57.1; ICE indel %: JDS246-Cas9 = 53.3; GeneArt CRISPR nuclease = 54.3, heiCas9 = 60.3; ICE knock-out score %: JDS246-Cas9 = 33.7; GeneArt CRISPR nuclease = 35.0, heiCas9 = 58.3. $R^2$> 0.9 (TIDE) and>0.9 (ICE) for all mRNAs tested. For representative indel spectrum for each mRNA, see *Figure 3—figure supplement 1*.

The online version of this article includes the following source data and figure supplement(s) for figure 3:

**Source data 1.** Raw data for scatter plot shown in *Figure 3*.

**Figure supplement 1.** Representative indel spectrum for each *Cas9* mRNA used in the cell culture assay.

ATTO-550-linked tracrRNA and the mRNAs of either *JDS246-Cas9*, *GeneArt CRISPR nuclease,* or *heiCas9*. The *Prx* locus was PCR amplified and sequenced. Similar to targeting *in organismo*, heiCas9 also exhibited the highest genome editing efficiency when compared to JDS246-Cas9 (TIDE: 123.6%, ICE: 113%) and GeneArt CRISPR nuclease (TIDE: 123.1%, ICE: 111%) in mammalian cell culture (*Figure 3*, *Figure 3—figure supplement 1*, $R^2 > 0.9$ (TIDE) and >0.9 (ICE) for all mRNAs tested). Notably, the KO-score efficiencies (ICE) amounted to 173% compared to JDS246-Cas9 and to 167% compared to GeneArt CRISPR nuclease, indicating higher abundance of frameshifts (*Hsiau et al., 2018*) at this genomic locus.

Remarkably, *heiCas9*-transfected cells showed a highly increased number of mutant alleles with an increased abundance of a 26 nt deletion when compared to GeneArt CRISPR nuclease and JDS246-Cas9 (*Figure 3—figure supplement 1*).

Given the observed boosting of Cas9 activity by the simple addition of the hei-tag, we next tested if the hei-tag also improves further Cas9-based techniques. Base editing is an increasingly applied method with a potential for therapeutics (*Antoniou et al., 2021*). Base editors are composed of a modified Cas9 that only nicks one DNA strand and does not introduce a double-strand break (Cas9 nickase or Cas9n) and a nucleotide deaminase for precisely targeted nucleotide editing (*Anzalone et al., 2020*). To increase the efficiency of base editors, several iterative rounds of optimization of the employed deaminases and linkers have been undertaken, yielding optimal performance with the newest variants (*Carrington et al., 2020*; *Cornean et al., 2022*; *Rosello et al., 2021*; *Zhao et al., 2020*). To investigate if the addition of the hei-tag provides an easy and straightforward alternative route for increasing the activity of a nuclear protein of interest, we selected a C-to-T base editor version with intermediate efficiency (BE4-Gam *Komor et al., 2017*) to introduce non-sense or severe miss-sense mutations into the pigmentation gene *oca2*. We employed our tool ACEofBASEs (*Cornean et al., 2022*) to design and evaluate sgRNA target sites that introduce non-synonymous codon mutations and/or pre-mature STOP codons upon editing of the respective open reading frame (ORF). We compared three different sgRNAs (*OlOca2 T1*, *T3*, and *T4*) employing the original BE4-Gam and the hei-tag fused variant (heiBE4-Gam). In the *oca2* ORF, the transition of cytosines 766, 922, and 997 to thymine all convert the respective codon to a pre-mature STOP (*OlOca2 T3*: C766T, leading to Q256*; *OlOca2 T4*: C922T, leading to Q308*; *OlOca2 T1*: C995-997T, leading to T332I and Q333*). Again, the loss of pigmentation was used as proxy for bi-allelic targeting efficiency following medaka one-cell stage injections with either one of the three sgRNAs (*OlOca2 T1*, *T3,* or *T4*, 30ng/µl) as well as 150ng/µl mRNA of either *BE4-Gam* or *heiBE4-Gam*. Screening and analysis was performed at 4.5 dpf as described above. For each sgRNA employed, heiBE4-Gam resulted in more pronounced loss of pigmentation in comparison to BE4-Gam (*Figure 4a*; control median = 0.0; medians BE4-Gam vs. heiBE4-Gam: *OlOca2 T1*, 0.6 vs. 28.0, p = 1.737e-20; *OlOca2 T3*, 0.0 vs. 0.8, p = 0.0471; *OlOca2 T4*, 93.8 vs. 170.1, p = 5.215e-12). Quantification of Sanger sequencing reads confirmed an increase of all C-to-T transitions at the *OlOca2 T1* target site when heiBE4-Gam was used (74.1% ± 8.9% for heiBE4-Gam vs. 44.2% ± 6.8% for BE4-Gam; *Figure 4—figure supplement 1*, three replicates containing five randomly picked embryos each). In particular, the C997T transition introducing a pre-mature STOP codon was increased 1.7-fold (i.e. 68% in heiBE4-Gam vs. 41% in BE4-Gam) in case of *heiBE4-Gam* (*Figure 4b and c*).

In conclusion, using the hei-tag to extend the ORFs of a mammalian codon-optimized *SpCas9* or a C-to-T base editor (BE4-Gam) severely enhanced the respective genome targeting efficiency.

## Discussion

While the use of the optimized NLS in the hei-tag explains the earlier and better performance of the hei-tagged versions of Cas9 and base editors in developing organisms, the impact of the specific topology of domains contained in the hei-tag remains elusive. It is speculated that the addition of certain peptide tags influences the efficacy and specificity of the fused protein of interest, due to their different isoelectric points and charge distributions (*Zhang et al., 2014*). Interestingly, our permutation screen demonstrated that although comprising the exact same peptides (for instance, compare MFO-Cas9-O [heiCas9] vs. OMF-Cas9-O and MSF-Cas9-S vs. SMF-Cas9-S in *Figure 1—figure supplement 1*), position of the particular tags relative to each other conveyed different genome editing efficiencies.

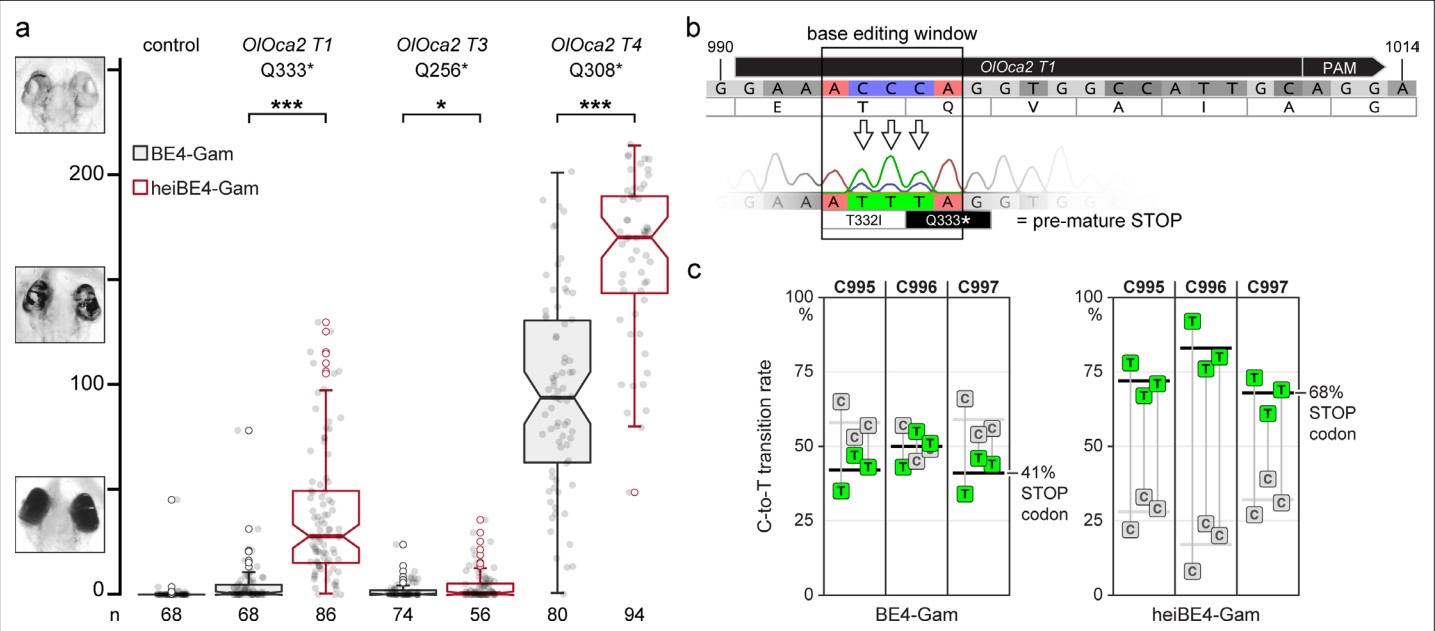

**Figure 4.** heiBE4-Gam mediates highly efficient cytosine-to-thymine (C-to-T) transitions in medaka embryos. Phenotypic range and quantification of heiBE4-Gam-mediated C-to-T transitions in medaka embryos. (**a**) Categories of typically observed loss-of-pigmentation phenotypes in *oca2* editants. The observed pigmentation phenotypes range from (almost) unpigmented eyes, that is, a very strong knock-out (top panel) over intermediate (central panel) to no loss of pigmentation (bottom panel). Quantification of phenotype resulting from injections with either *BE-Gam* or *heiBE-Gam* (red) mRNA and *OlOca2 T1*, *T3*, or *T4* sgRNAs. Note: dramatic increase of bi-allelic knock-out rate when using *heiBE-Gam*. n, number of eyes analyzed. Control median = 0.0; medians BE4-Gam vs. heiBE4-Gam: *OlOca2 T1*, 0.6 vs. 28.0, p = 1.737; *OlOca2 T3*, 0.0 vs. 0.8, p = 0.0471; *OlOca2 T4*, 93.8 vs. 170.1, p = 5.215e-12. Bold lines, median values. Statistical analysis performed in R, pairwise Wilcoxon rank sum test. (**b**) Schematic representation of base editing window in *OlOca2 T1* target site (PAM, protospacer adjacent motif). C-to-T transition of C995 and C996 edits the threonine (**T**) codon to isoleucine (**I**) (T332I); C997T creates a pre-mature STOP codon (Q333*). Nucleotide positions refer to the *oca2* open reading frame. (**c**) Quantification of Sanger sequencing reads at nucleotides C995, C996, C997 inside the base editing window of three injected embryo pools (five embryos each) reveals overall dramatic increase of C-to-T base transition when using heiBE4-Gam. Note 1.7-fold increase of C997T transition, that is, efficient introduction of a pre-mature STOP codon. Mean values indicated by bold horizontal lines, *Figure 4—figure supplement 1*.

The online version of this article includes the following source data and figure supplement(s) for figure 4:

**Source data 1.** Raw data for quantifications shown in *Figure 4a*.

**Figure supplement 1.** Increased cytosine-to-thymine (C-to-T) transition in medaka embryo pools injected with *heiBE4-Gam*.

The hei-tag renders the resulting heiCas9 into a highly efficient endonuclease with broad applicability overcoming the limitations of current *SpCas9* variants by dramatically increasing the efficiency of targeted genome editing *in organismo*, as demonstrated in two evolutionarily distant fish models, as well as in mouse cell culture. In those systems, heiCas9 leads to a high abundance of identical mutant alleles, important for testing specific hypotheses or introducing site-specific modifications by homology-directed repair (*Gutierrez-Triana et al., 2018*). Conversely, Cas9 variants without the hei-tag are better suited for targeted screening approaches since they introduce a large number of different mutant alleles. heiCas9 markedly increased the (bi-allelic) targeting rate alongside a decrease in allele variance, indicating a high targeting efficiency already at the earliest stages of development. Precedentially such early targeting in developing organisms was most of all reported using RNPs (*Kroll et al., 2021*; *Wu et al., 2018*), yet mRNA injection of heiCas9 is fully comparable to these protein approaches. The benefits of using mRNA over protein are apparent: new Cas9 variants can easily be generated and produced cost-efficiently by highly reproducible in vitro transcription, a standard method in molecular biology labs.

In light of the ever-expanding CRISPR tool kit, the addition of the hei-tag provides the means to boost current specialized and future variants, as the simple addition of the hei-tag sequence also potentiated the activity of a cytosine base editor, with heiBE4-Gam resulting in an overall increase of about 30% of C-to-T transition rates (*Figure 4* and *Figure 4—figure supplement 1*). Taken together, the boosting activity of the hei-tag is neither limited by the species nor the approach, making it

**Table 1.** Primer sequences used for Cas9 variant cloning.

Restriction enzyme sites used for cloning are indicated in italics (*AgeI* in the forward primer, *XbaI* in the reverse primer), underscored sequence, binding to Cas9 open reading frame (ORF). F, flexible linker; I, internal linker; M, cMyc-tag; O, optimized NLS (*Inoue et al., 2016*); S, SV40 NLS (*Kalderon et al., 1984*); Xl, bipartite *Xenopus laevis* nucleoplasmin NLS (*Dingwall et al., 1988*). For instance, to establish the heiCas9 ORF, primers MFO-Cas9_fwd and Cas9-O_rev were used.

| Primer name | Primer sequences in 5'–3' |
| --- | --- |
| MFO-Cas9_fwd | AATTTACCGGTTTACCATGGAGCAGAAGCTGATCAGCGAGGAGGACCTGGGAGGAAGCGGACCACCTCCCAAGAGGCCCAGGCTGGACCTCGAG<u>GATAAAAAGTATTCTATTGGTTTAG</u> |
| MIS-Cas9_fwd | AATTTACCGGTTTACCATGGAGCAGAAGCTGATCAGCGAGGAGGACCTGGGTATCCACGGAGTCCCAGCAGCCGCTCCAAAGAAGAAGCGTAAGGTA<u>GATAAAAAGTATTCTATTGGTTTAG</u> |
| MSF-Cas9_fwd | AATTTACCGGTTTACCATGGAGCAGAAGCTGATCAGCGAGGAGGACCTGATGGCTCCAAAGAAGAAGCGTAAGGTAGGAGGAAGCGGA<u>GATAAAAAGTATTCTATTGGTTTAG</u> |
| OMF-Cas9_fwd | AATTT*ACCGGT*TTACCATGCCACCTCCCAAGAGGCCCAGGCTGGACCTCGAGGAGCAGAAGCTGATCAGCGAGGAGGACCTGGGAGGAAGCGGA<u>GATAAAAAGTATTCTATTGGTTTAG</u> |
| SMF-Cas9_fwd | AATTTACCGGTTTACCATGGCTCCAAAGAAGAAGCGTAAGGTACTCGAGGAGCAGAAGCTGATCAGCGAGGAGGACCTGGGAGGAAGCGGA<u>GATAAAAAGTATTCTATTGGTTTAG</u> |
| Cas9-O_rev | AATTT*TCTAGA*TTAGTCCAGCCTGGGCCTCTTGGGAGGAGGGGATCC<u>GTCACCCCCAAGCTGTGAC</u> |
| Cas9-S_rev | AATTT*TCTAGA*TTAATCTACCTTACGCTTCTTCTTTGGAGCAGCGGATCC<u>GTCACCCCCAAGCTGTGACA</u> |
| myc-Cas9_fwd | AATTT*ACCGGT*CAAACATGGAGCAGAAGCTGATCAGCGAGGAGGACCTGATGGCCCCAAAGAAGAAGCGGAAGGTC |
| myc-Cas9_rev | AATTT*TCTAGA*TTACTTTTTCTTTTTTGCCTGGCCGGC |

a powerful tweak to swiftly upgrade any specifically adapted Cas-based genome editing approach (*Anzalone et al., 2020*).

# Materials and methods

## Fish maintenance

Zebrafish (*D. rerio*) and medaka (*O. latipes*) fish were bred and maintained as previously described (*Koster et al., 1997*; *Westerfield, 2000*). The animal strains used in the present study were zebrafish AB/back and medaka Cab. All experimental procedures were performed according to the guidelines of the German animal welfare law and approved by the local government (Tierschutzgesetz §11, Abs. 1, Nr. 1, husbandry permit number 35-9185.64/BH Wittbrodt).

## Cloning of Cas9 variants

The mammalian codon-optimized (Geneious 8.1.9, https://www.geneious.com) *Cas9* sequence was gene-synthesized (GeneArt, ThermoFisher Scientific) as template for cloning the permutated peptide-tag Cas9 fusions (*Supplementary file 2*) using primers (*Table 1*) containing the sequences coding for a myc-tag (EQKLISEEDL), flexible or internal linkers and an SV40 (PKKKRKV) or optimized oNLS (PPPKRPRLD) (*Inoue et al., 2016*; *Figure 1—figure supplement 1*). Cloning into the pCS2+ plasmid (*Rupp et al., 1994*) (multiple cloning site extended for *AgeI* site downstream of *BamHI* site) was performed using *AgeI* and *XbaI* restriction sites included in the 5' region of the forward or reverse primers, respectively. See *Supplementary file 1* for full sequence of *heiCas9*. For consistent mRNA synthesis, the published *myc-Cas9* (*Zhang et al., 2014*) (MSI-Cas9-Xl) was re-established with the pX330-U6-Chimeric_BB-CBh-hSpCas9 vector as template, primer-based exchange of the N-terminal FLAG tag with the myc-tag sequence and brought into pCS2+ (*Rupp et al., 1994*) using *AgeI* and

**Table 2.** Primer sequences used for BE4-Gam and heiBE4-Gam cloning.

| Primer name | Primer sequences in 5'–3' |
| --- | --- |
| pCS2+_backbone_fwd | GCCTCTAGAACTATAGTGAGTCG |
| pCS2+_backbone_rev | ATGGGATCCTGCAAAAAGAACAAG |
| hei-tag_fragment_fwd | CTTGTTCTTTTTGCAGGATCCCATTTACCATGGAGCAGAAGCTG |
| hei-tag_fragment_rev | GCTGGTTTAGCCTCGAGGTCCAGCCTGG |
| Gam_Mu-APOBEC1-Cas9n_fragment_fwd | GACCTCGAGGCTAAACCAGCAAAACGTATCAAG |
| Gam_Mu-APOBEC1-Cas9n_fragment_rev | CTAGGGCCTTGAGAAGTGTC |
| Cas9n-UGI_fragment_fwd | GACACTTCTCAAGGCCCTAG |
| Cas9n-UGI_fragment_rev | CAGAGTCACCCCCAAGCTG |
| 2xUGI-oNLS_fwd | CAGCTTGGGGGTGACTCTG |
| 2xUGI-oNLS_rev | CGACTCACTATAGTTCTAGAGGCTTAGTCCAGCCTGGGCCTCTTGGGAGGGGGAGAACCACCAGAGAGC |

*XbaI* restriction sites included in the 5' region of the respective primers as well. pX330-U6-Chimeric_BB-CBh-hSpCas9 was a gift from Feng Zhang (Addgene plasmid #42230) (*Cong et al., 2013*).

## Cloning of BE4-Gam and heiBE4-Gam

BE4-Gam was subcloned from pCMV(BE4-Gam) (Addgene plasmid #100806, a gift from David Liu) (*Komor et al., 2017*) in a two-step process, first into pJET1.2 (Thermo Scientific), then into pGGEV4 (*Kirchmaier et al., 2013*) (Addgene plasmid #49284), by *BamHI, EcoRV,* and *KpnI* restriction sites to create pGGEV4(BE4-Gam). heiBE4-Gam was assembled into pCS2+ (*Rupp et al., 1994*) by NEBuilder HiFi DNA Assembly (NEB) with four inserts using Q5 polymerase PCR products (NEB): pCS2+ backbone, hei-tag fragment, Gam Mu-APOBEC1-Cas9n fragment, Cas9n-UGI fragment, 2xUGI-oNLS (see *Table 2* for primers used).

## sgRNA design

All *sgRNAs* for medaka (*OlOca2, Rx2, Rx3, Cryaa*) and zebrafish (*DrOca2*) were designed using the CCTop target predictor with standard parameters (*Stemmer et al., 2015*). The sgRNAs used for base editing (*OlOca2 T1, T3, T4*) were designed or evaluated using ACEofBASEs (*Cornean et al., 2022*) and selected for introducing a pre-mature STOP codon. The following target sites were used [PAM in brackets]: *OlOca2 T1* (GAAACCCAGGTGGCCATTGC[AGG]), *OlOca2 T2* (TTGCAGGAATCATTCTGTGT[GGG]), *OlOca2 T3* (GATCCAAGTGGAGCAGACTG[AGG]), *OlOca2 T4* (CACAATCCAGGCCTTCCTGC[AGG]) *DrOca2 T1* (GTACAGCGACTGGTTAGTCA[TGG]), *DrOca2 T2* (TAAGCACGTAGACTCCTGCC[AGG]), *Rx2* (GCATTTGTCAATGGATACCC[TGG]), *Cryaa* (GGGAGAAGTGCTTGACATCC[AGG]), *Rx3* (AGCAGAGCGCGCAAAGAACC[AGG]). *OlOca2 T1, OlOca2 T2,* and *DrOca2 T1* were the same as in *Hammouda et al., 2019*, *OlOca2 T3* was the same as in *Lischik et al., 2019* (OCA2_4), *OlRx2* and *OlCryaa* are from *Stemmer et al., 2015*, and *OlRx3* is the same used in *Zilova et al., 2021*. Cloning of sgRNA templates was performed as described (*Stemmer et al., 2015*). Plasmid DR274 was a gift from Keith Joung (Addgene plasmid #42250) (*Hwang et al., 2013*).

In vitro transcription of mRNA pCS2+ constructs in this work were linearized using NotI-HF (NEB) except for zCas9 – linearized with HpaI (NEB). The pGGEV4(BE4-Gam) was linearized using SpeI-HF (NEB). mRNA was transcribed in vitro using the mMESSAGE mMACHINE SP6 transcription kit (ThermoFisher Scientific, AM1340). pCS2-nCas9n (zCas9) was a gift from Wenbiao Chen (Addgene plasmid #47929) (*Jao et al., 2013*). The *JDS246-Cas9* was linearized with MssI FD (ThermoFisher Scientific) and transcribed in vitro using the mMESSAGE mMACHINE T7 Ultra Transcription Kit (ThermoFisher Scientific, AM1345). JDS246-Cas9 was a gift from Keith Joung (Addgene plasmid #43861). sgRNAs were synthesized using the MEGAscript T7 transcription kit (ThermoFisher Scientific, AM1334) after plasmid digestion with DraI FD (ThermoFisher Scientific).

## Microinjection

All microinjections were performed at the one-cell stage. At standard concentrations, zebrafish and medaka zygotes were injected with 150 ng/µl Cas9 (variant) mRNA, *Oca2* sgRNAs at 30 ng/µl, and *H2B-GFP* mRNA at 10 ng/µl as injection tracer. The multiplexing injection mixes contained 12.5 ng/µl per sgRNA (*OlOca2 T2*, *Rx2*, *Rx3*, *Cryaa*) and 15 ng/µl of either *zCas9* or *heiCas9* mRNA as well as 20 ng/µl mCherry mRNA as injection tracer. For the protein injections, 24 µM RNP mix (*Kroll et al., 2021*) was assembled in Cas9 buffer (20 mM Tris-HCl, 600 mM KCl, 20% glycerol; *Wu et al., 2018*) by mixing 61 µM Alt-R S.p. Cas9 Nuclease V3 (IDT) with 5710 ng of each sgRNA *OlOca2 T1* and *T2*; 285.6 ng GFP mRNA were added as injection tracer. The mix was incubated for 5 min at 37°C and stored on ice until further dilution and injection. To obtain 5 µM RNPs (*Wu et al., 2018*), the 24 µM RNP mix was diluted in a 1:1 mixture of Cas9 buffer and nuclease-free water. Five µM RNP solution was further diluted in a 1:1 mixture of Cas9 buffer and nuclease-free water to obtain 1.765 µM RNPs.

For the base editing experiments, medaka zygotes were injected with *BE4-Gam* or *heiBE4-Gam* mRNA at 150 ng/µl, respective *Oca2* sgRNA at 30 ng/µl, and *GFP* mRNA at 20 ng/µl as injection tracer. All injected embryos were maintained at 28°C in zebrafish medium (*Westerfield, 2000*) or medaka embryo rearing medium (ERM, 17 mM NaCl; 40 mM KCl; 0.27 mM CaCl$_2$•2H$_2$O; 0.66 mM MgSO$_4$•7H$_2$O, 17 mM HEPES).

Embryos were screened for *GFP* or *mCherry* expression 4–7 hr or 1 day after injection using a Nikon SMZ18 stereomicroscope, and uninjected specimens were discarded.

## Image acquisition and phenotype analysis

Medaka 4.5 dpf embryos (*Iwamatsu, 2004*) and zebrafish 2.5 dpf (*Kimmel et al., 1995*) embryos were fixed with 4% paraformaldehyde in 2× PTW (2× PBS pH 7.3, 0.1% Tween 20). Images of medaka embryos were acquired with the high content screening ACQUIFER Imaging Machine (DITABIS AG, Pforzheim, Germany). Images of zebrafish embryos were acquired with a Nikon digital sight DS-Ri1 camera mounted onto a Nikon Microscope SMZ18 and the Nikon Software NIS-Elements F version 4.0. Only properly developed embryos were included in the following analysis. Image analysis was performed with Fiji (*Schindelin et al., 2012*), that is, mean gray values were obtained on minimum intensity projections and locally thresholded (Phansalkar algorithm with parameters r = 20, p = 0.4, k = 0.4) pictures and elliptical selections for each individual eye. The mean gray value per eye was used for the boxplot and statistical analysis (pairwise comparisons using Wilcoxon rank sum test, Bonferroni corrected) in RStudio (*Team, 2020*).

## Targeted amplicon sequencing via illumina

The multiplex approach was genotyped on DNA extractions of pools with each replicate containing eight randomly picked crispants per *zCas9* or *heiCas9* injection or six control specimens. DNA was prepared by grinding and lysis in DNA extraction buffer (0.4 M Tris/HCl pH 8.0, 0.15 M NaCl, 0.1% SDS, 5 mM EDTA, pH 8.0, 1 mg/ml proteinase K) at 60°C overnight. Proteinase K was inactivated at

**Table 3.** Locus-specific primers with 5' partial illumina adapter sequences.

Locus-specific primers with Illumina adapter sequence underscored.

| Primer name | Primer sequences in 5'–3' |
| --- | --- |
| oca2_F | ACACTCTTTCCCTACACGACGCTCTTCCGATCTCGTTAGAGTGGTATGGAGAACTGT |
| oca2_R | GACTGGAGTTCAGACGTGTGCTCTTCCGATCTATGGTCCTCACATCAGCAGC |
| cryaa_F | *ACACTCTTTCCCTACACGACGCTCTTCCGATCT*CGCCATTTGCTTGTGTGTCA |
| cryaa_R | GACTGGAGTTCAGACGTGTGCTCTTCCGATCTAGTCTAGGAGGATGGGGCAG |
| rx2_F | ACACTCTTTCCCTACACGACGCTCTTCCGATCTAGAGGCACAAGAACTATTTGTTGATC |
| rx2_R | GACTGGAGTTCAGACGTGTGCTCTTCCGATCTAGGGCTCCGTTAACTTTGGG |
| rx3_F | ACACTCTTTCCCTACACGACGCTCTTCCGATCTATGCAAACCAAGAAAGCGCC |
| rx3_R | GACTGGAGTTCAGACGTGTGCTCTTCCGATCTTGGGATTTCTCAAAGGCCCG |

95°C for 10 min and the solution was diluted 1:2 with nuclease-free water. For each DNA extraction, small libraries were constructed by PCR amplifying the four regions of interest (295–362 bp, *OlOca2*, *rx2*, *rx3*, *cryaa*) using locus-specific primers with 5' partial illumina adapter sequences (*Table 3*) and Q5 Hot Start High-Fidelity DNA Polymerase (New England Biolabs). PCR products were run on a 1% agarose gel, respective bands were excised and cleaned up using the Monarch DNA Gel Extraction Kit (New England Biolabs). PCR products from the same genomic DNA source were pooled to equimolarity at 25 ng/µl and submitted to GeneWiz (Azenta Life Sciences) for sequencing (Amplicon-EZ: Illumina MiSeq, 2 × 250 bp sequencing, paired-end) obtaining between 48,018 and 96,899 reads per sample.

## Analysis and plotting of next-generation sequencing data

Amplicon sequencing data was analyzed with CRISPResso2 v.2.1.2 (*Clement et al., 2019*) using the default -n nhej parameters. Demultiplexing was achieved by mapping to the four different wild-type loci, respectively. Downstream analysis was conducted using R v.3.6.3 in R studio (*Team, 2020*) (package: ggplot2 *Wickham, 2016*), with data sourced from 'CRISPResso_quantification_of_editing_frequency.txt' output table. To determine the average read count per modified allele, the 'Alleles_frequency_table.txt' output table was used. The number of modified alleles was determined by filtering > 'Read_status' > modified. Average read count per modified allele = modified reads/N modified alleles.

## Genotyping of editants

Genotyping was performed on DNA extractions (see above) of three replicates containing five randomly picked editants each of BE4-Gam and heiBE4-Gam injections. Q5 polymerase (NEB), primers fwd 5'-GTTAAAACAGTTTCTTAAAAAGAACAGGA-3' and rev 5'-AGCAGAAGAAATGACTCAACATTTTG-3' (annealing at 62°C) were used on 1 µl of diluted DNA sample according to the manufacturer's instructions with 30× PCR cycles. PCR products were analyzed on a 1% agarose gel, bands excised, DNA extraction performed using innuPREP Gel Extraction Kit (Analytik Jena) according to the manufacturer's instructions and subjected to Sanger sequencing (see below).

## Cell lines

Mouse SW10 cells (ATCC, CRL-2766, Lot number 4117643) were cultured in DMEM (Gibco) supplemented with 1 g/ml glucose containing 10% FCS (Sigma), 1% penicillin (10,000 units/ml; Gibco), and 1% streptomycin (10 mg/ml; Gibco) and maintained at 33°C and 5% $CO_2$ and regularly tested negative for mycoplasma infections. Cells were passaged at 80–90% confluency. Twenty-four hr before transfection cells were seeded in a density of 85,000 cells per 12 wells.

CRISPR Transfection crRNA targeting exon 6 (TCGTATCCAGACACCGTCCC[GGG], PAM in brackets) of the mouse *Periaxin* (*MmPrx*) gene was selected from the IDT (crRNA XT) predesign crRNA database. crRNA (50 µM) and Alt-R CRISPR-Cas9 tracrRNA, ATTO-550 (50 µM; IDT, 1075927) were diluted in nuclease-free duplex buffer (IDT) to a final concentration of 1 µM and incubated at 95°C for 5 min. One µg of the corresponding *Cas9* mRNA (*GeneArt CRISPR nuclease* Invitrogen, A29378; *JDS246-Cas9* or *heiCas9*) and 15 µl of tracrRNA+crRNA Mix (1 µM) were diluted in 34 µl Opti-MEM I (Gibco) and mixed with 3 µl Lipofectamine RNAiMAX (ThermoFisher) diluted in 47 µl Opti-MEM I. The tracrRNA+crRNA lipofection mix was incubated for 20 min at RT. Cell culture medium was exchanged with 900 µl Opti-MEM I and the tracrRNA+crRNA lipofection mix was added dropwise to the well. After 48 hr, genomic DNA was extracted using the DNeasy Blood and Tissue Kit (Qiagen, 69506) following the manufacturer's protocol. Q5-PCR was carried out using primers flanking the targeted exon 6 (fwd 5'-GAGACACTCACTCCAGACCC-3'; rev 5'-ACTCAGTAACCCAACAGCCA-3') and 30 cycles. PCR amplicons were purified using the Monarch DNA Gel Extraction Kit (NEB, T1020S) and subjected to sequencing.

## Sanger sequencing

Sanger sequencing was performed by Eurofins Genomics using fwd 5'-GTTAAAACAGTTTCTTAAAAAGAACAGGA-3' to evaluate base editing at *OlOca2 T1* target site and using fwd 5'-GAGACACTCACTCCAGACCC-3' and rev 5'-ACTCAGTAACCCAACAGCCA-3' to evaluate genome editing of the *Prx* locus in SW10 cells. Quantification of base editing from Sanger sequencing reads was performed

with EditR (*Kluesner et al., 2018*). Genome editing efficiency was assessed by sequence analysis using the TIDE web tool (*Brinkman et al., 2014*) and by ICE (*Hsiau et al., 2018*) using default parameters and indel size range up to 30 bp.

## Data visualization

Data visualization and figure assembly was performed using Fiji (*Schindelin et al., 2012*), ggplot2 (*Wickham, 2016*) in RStudio (*Team, 2020*), Geneious Prime 2019.2.1, Adobe Illustrator CS6 and Affinity Designer 1.10.5.

## Acknowledgements

This research was supported by grants of the European Research Council (ERC-SyG H2020, 810172), the Excellence Cluster '3D Matter Made to Order' (3DMM2O, EXC 2082/1 Wittbrodt C3) funded through the German Excellence Strategy via Deutsche Forschungsgemeinschaft (DFG), CRC873, project A3 and FOR2509 project 10 (WI 1824/9-1) to JW and CRC1118, project S03 to MF. AC, BW, RM, and TT-S are members/alumni of HBIGS, the Heidelberg Biosciences International Graduate School. BW was supported by the Deutsches Zentrum für Herz-Kreislauf-Forschung (DZHK B20-024 SE). We thank T Kellner for sgRNA, base editor, and Cas9 mRNA synthesis. We are thankful to M Majewski, E Leist, S Erny and A Saraceno for fish husbandry. We thank all members of the Wittbrodt lab for their critical, constructive feedback on the procedure and the manuscript.

## Additional information

### Competing interests

Thomas Thumberger, Tinatini Tavhelidse-Suck, Jose Arturo Gutierrez-Triana, Joachim Wittbrodt: patent application pending (EP21166099.8) related to the findings described. The other authors declare that no competing interests exist.

### Funding

| Funder | Grant reference number | Author |
| --- | --- | --- |
| Deutsche Forschungsgemeinschaft | CRC873 project A3 | Joachim Wittbrodt |
| Deutsche Forschungsgemeinschaft | FOR2509 P10 WI 1824/9-1 | Joachim Wittbrodt |
| Deutsche Forschungsgemeinschaft | CRC1118 project S03 | Marc Freichel |
| H2020 European Research Council | 810172 | Joachim Wittbrodt |
| Deutsche Forschungsgemeinschaft | 3DMM2O, EXC 2082/1 Wittbrodt C3 | Joachim Wittbrodt |

The funders had no role in study design, data collection and interpretation, or the decision to submit the work for publication.

### Author contributions

Thomas Thumberger, Conceptualization, Data curation, Formal analysis, Investigation, Software, Validation, Visualization, Writing – original draft, Writing – review and editing; Tinatini Tavhelidse-Suck, Conceptualization, Data curation, Formal analysis, Investigation, Validation, Writing – original draft, Writing – review and editing; Jose Arturo Gutierrez-Triana, Conceptualization, Formal analysis, Investigation, Validation, Writing – original draft, Writing – review and editing; Alex Cornean, Data curation, Formal analysis, Investigation; Rebekka Medert, Formal analysis, Investigation; Bettina Welz, Investigation; Marc Freichel, Resources, Supervision; Joachim Wittbrodt, Conceptualization, Funding acquisition, Project administration, Resources, Supervision, Writing – original draft, Writing – review and editing

## Author ORCIDs
Thomas Thumberger  http://orcid.org/0000-0001-8485-457X
Tinatini Tavhelidse-Suck  http://orcid.org/0000-0002-6103-9019
Alex Cornean  http://orcid.org/0000-0003-3727-7057
Marc Freichel  http://orcid.org/0000-0003-1387-2636
Joachim Wittbrodt  http://orcid.org/0000-0001-8550-7377

## Decision letter and Author response
Decision letter https://doi.org/10.7554/eLife.70558.sa1
Author response https://doi.org/10.7554/eLife.70558.sa2

---

## Additional files

### Supplementary files
• Supplementary file 1. Nucleotide and translated amino acid sequence of heiCas9.

• Supplementary file 2. Sequences of peptide tags fused to mammalian SpCas9 (*Figure 1*, *Supplementary file 1*). M, cMyc-tag; O, optimized nuclear localization signal (NLS) (*Inoue et al., 2016*); S, SV40 NLS (*Kalderon et al., 1984*); Xl, bipartite *Xenopus laevis* nucleoplasmin NLS (*Dingwall et al., 1988*).

• Supplementary file 3. Allele variants and abundance in *OlOca2, rx2, rx3,* and *cryaa*. Sequences and abundance of all locus-mapped reads per replicate (pool) and locus (*OlOca2, rx2, rx3, cryaa*) of multiplexing with either *zCas9* or *heiCas9* mRNA injections (*Figure 2*). Sequences of allele variants (with more than 100 reads) displayed.

• Transparent reporting form

### Data availability
All data generated or analysed during this study are included in the manuscript and supporting files.

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
