## [Editor Report]

This manuscript describes the addition of a short tag on the Cas9 nuclease as a means to improve genome editing efficiency. Importantly, the authors have tested their approach on several genomic targets, model organisms, and Cas9 derivative engineering tools. Overall, these findings support the possible general applicability of this tag for improving the outcomes of a wide range of modern Cas9 based applications, including Base Editors. Adding to a recent report of improving Prime Editing by optimising the NLS, this paper reinforces the notion that there is still unexplored space in altering genome engineering activity in a modular way.

---

## [Decision Letter]

**Decision letter after peer review:**

Thank you for submitting your article "hei-tag: a highly efficient tag to boost targeted genome editing" for consideration by *eLife*. Your article has been reviewed by 3 peer reviewers, including Zacharias Kontarakis as Reviewing Editor and Reviewer #1, and the evaluation has been overseen by Didier Stainier as the Senior Editor. The reviewers have opted to remain anonymous.

Essential revisions:

1) Benchmark hei-tag activity against the state-of-art genome engineering tool in each experiment (see reviewer comments – also comparison to RNP use in fish is highly desirable).

2) Provide heiCas9 editing data across multiple targets/gRNAs, ideally showing diverse characteristics (e.g. genomic location, sequence, activity).

3) Edit manuscript to clearly describe the strengths and weaknesses of the "hei-tagging" method.

*Reviewer #1 (Recommendations for the authors):*

I have a few concerns regarding the way the authors structure and present their claims:

1) One of the main take home messages (based on the Abstract) is that the hei-tag boosts the activity of a "wide variety" of genome editing tools. Even though the BE data are promising, the manuscript could use more examples (e.g. prime editors, that have been used in zebrafish – Petri et al., 2021). Can the hei-tag make editing possible in cases where it is undetectable with the state of the art tools?

2) Throughout the manuscript, the authors use only a handful of guides. Adding more examples, especially using guides that show a range of efficiencies with the non-hei-tag Cas9 would provide a better description of the strengths and limitations of the hei-tag.

3) What is the rationale of using the Cas9 from Hwang et al., and not the zebrafish codon optimised Cas9 from Jao et al?

4) Why do the authors use the GeneArt nuclease as a "state of the art" for the cell culture? What other tags are included in this nuclease? We can see that the worse score in Fig1d comes from JDS246. Not having the hei-tag is only one difference between JDS246 and heiCas9. One other is that heiCas9 lacks the FLAG (Myc-Cas performs already pretty well, however zebrafish data are not shown). How does GeneArt Cas9 compare in that respect? Such information should be presented to readers.

5) In Figure 1, what do the single guide results look like? Boosting deletion outcomes using two guides is not the same as boosting activity of single guide editing. Is the activity of both guides improved? Are they just being brought at the same level? Did the authors use sequencing to analyse what happened at the DNA level?

6) The hei-tag could be affecting editing outcomes, rather than ON target activity. In Figure S1, it is clear that the frequency of the in frame -15 allele is reduced, while overall efficiency in not dramatically affected. Changing the editing profile by using different nucleases (variants) is not the same as boosting efficiency in KO. The in-frame indels could be favored in other sites, thus reducing KO-score (if out of frame are regarded as desired outcome).

7) Generally, it is advised to use NGS data for strong claims about editing efficiency.

8) If ON-target activity is boosted, increased OFF-target activity is of concern, especially in cell culture when editing accuracy and precision are key. Since the main focus of the authors is not mammalian cell culture or therapeutic applications, they should at least be clear about this limitation.

Generally, the manuscript would be strengthened by including more examples of editing tools (desirable, but not absolutely required), include analyses of more guides (required), add sequencing data wherever possible (highly advised), adequately discussing strengths and limitations of the hei-tag (required), presenting some NGS data (recommended), toning down some "generalised" claims (unless more supporting data are provided).

*Reviewer #2 (Recommendations for the authors):*

Several reports in zebrafish have shown that Cas9/gRNA complexes injections can reach gene editing up to 100% in F0 embryos including when targeting multiple loci at the same time (See for instance Wu et al., 2018 and Kroll et al., 2021). In light of these impressive results the utility of the present method for the fish community is overstated at best without mentioning these points anywhere in the discussion.

The authors should attempt to compare the injection of heiCAS9 protein as this is the current preferred and most efficient method in the fish community.

In the present manuscript is not clear what is the purpose of the Myc tag in the construct. This is not used at all to visualize protein localization or purification. Do the authors believe that the addition of Myc increases Cas9 activity? If not giving that this is a method paper they should demonstrate the utility in this specific contest.

The zebrafish experiment lacks the comparison of the heicas9 with the most efficient myc-Cas9 leading to an overestimation to the improved efficiency of the construct.

Finally methodological papers based on a single locus are difficult to appreciate as they may be influenced by "the gambler's luck" (i.e. the chosen locus cold be a fortunate pick). These results should be extended to other genes as it's standard in the field (see the two publications above).

Finally the base editor experiment are equally strongly biased. The comparison with BE4-Gam is not representative of the current state of the art were several reports using ancBE4Max (See for instance Carrington et al., 2020, Zao et al., 2020 and Rosello et al., 2021) show highly improved results. In lights of these papers the statement "Notably, in heiBE4-Gam injections, for each of the three cytosines in the base editing window, the C-to-T transition rate was higher than 60%, a level never observed in BE4-Gam injected embryos" is not true as similar or higher level of C to T conversion have been reported. Again this a comparison with ancBE4Max should be extended to multiple loci.

*Reviewer #3 (Recommendations for the authors):*

1. In the abstract, the authors introduced factors that leave room for the improvement of gene editing efficiency in CRISPR/Cas9 tools: (1) nuclear localization signal -citing Cong et al., 2013; and (2) protein tags for "immediate detection or straight-forward purification" and linkers to "avoid steric hinderance impacting on activity" -citing Zhang et al., 2014. However, such conclusions were never made in either of the originally cited papers. Cong et al., did not compare the editing activity with and without a NLS signal. On the contrary, there are partial evidence indicating that Cas9 protein may not require an NLS to assist import into the nucleus (Hu et al., G3, 2018). In Zhang et al., it was suspected that the addition of a flag or myc tag changed the charge distribution of the Cas9 protein, thus increasing its specificity and efficacy. No statement was made about the purification or the linker. It is OK to introduce the relevant background, but I find it problematic to misinterpret the cited literature to show conclusions they did not make.

2. The authors used modified version of NLS in the hei-tag construct to facilitate early nuclear targeting, while a straight-forward way to make nuclear-targeted Cas9 available in the cell is to directly inject the (nuclear-targeting) Cas9 protein. The authors should either provide this control experiment, or clarify why they only chose to build RNA-based systems.

3. It is widely known that the gRNA design is a critical factor affecting the gene editing efficiency. While hei-tag shows an increased bi-allelic editing efficiency than the control constructs, it is not clear whether this boost is universal with different targeted genes and different sgRNA designs.

4. JDS246-Cas9 was chosen as the baseline construct to evaluate the boost of editing efficiency. Given this was a construct originally made for mammalian cells, it is not clear whether it represents the state-of-art editing techniques, especially in zebrafish. While it is unrealistic to test all the available tools, other systems have been reported with high bi-allelic editing efficiency specifically in zebrafish should be introduced as a control (e.g. Jao et al., 2013, PNAS -disclaimer: the reviewer was not a maker of this tool).

5. In the base-editing experiments, the injected fish showed various level of eye-pigmentation colors, in contrast to the knockout experiment where cells devoid of pigment appear in patches on the eye. The authors should provide an explanation of why this is the case and justify why the pigmentation level has to be quantified differently in Figure 3 than Figure 1.

---

## [Author Response]

Essential revisions:1) Benchmark hei-tag activity against the state-of-art genome engineering tool in each experiment (see reviewer comments – also comparison to RNP use in fish is highly desirable).

In the revised version of the manuscript we now extensively compare the hei-tagged Cas9 (heiCas9) against state-of-the-art Cas variants, including RNPs.

Strikingly, heiCas9 clearly outperforms the state-of-the-art Cas variants and RNPs. Only when applied at highest concentrations (highly viscous injection mix) RNPs show a slightly increased activity. Following the reviewer’s suggestion, we now use the zebrafish codon optimized nls-zCas9-nls from Jao et al., 2013 as reference in both, medaka and zebrafish. heiCas9 mRNA in both cases massively outperforms the reference. This is now presented in the new Figure 1.

Given that all those comparisons were performed at saturating conditions in the plateau of Cas activity, we subsequently diluted the concentrations of the respective Cas variants by a factor of 10 and compared their activity in the exponential phase for four different target loci in a multiplexing approach. Here, the massive performance boost provided by the hei-tag becomes fully apparent, which was partially masked at saturating conditions.

We quantified the results by Illumina sequencing of the respective multiplexed loci and show a high specific activity of the heiCas9 in contrast to the zCas9. Strikingly, the allele variance per locus is clearly reduced in the heiCas9 experiment (even under multiplexing conditions) in comparison to the state-of-the-art Cas9 variant. This argues for an early (in development) and precise activity of heiCas9.

Those findings are now presented in the all new Figure 2 in the revised version of the manuscript.

2) Provide heiCas9 editing data across multiple targets/gRNAs, ideally showing diverse characteristics (e.g. genomic location, sequence, activity).

To address this point, we have been comparing the activity of the state-of-the-art Cas9 variant (zCas9) with heiCas9 on four different genomic loci (*oca2*, *rx2*, *rx3*, *cryaa*) employing sgRNAs with different activities in a multiplexing approach.

Editing efficiency on the respective loci in this multiplexing experiment was quantified by Illumina sequencing.

This analysis revealed a different editing efficiency for the different loci of both, the nls-zCas9-nls (Jao et al., 2013) as well as the heiCas9. In all cases, however, heiCas9 performed better than the state-of-the-art Cas9 variant. Interestingly, besides a massive increase in editing events, heiCas9 editing resulted in a clear drop of the allele variance. This indicates that heiCas9 edited more efficiently and acted earlier in development (than the zCas9) resulting in a lower number of different alleles. This is now presented in a new Figure 2 in the revised version of the manuscript.

3) Edit manuscript to clearly describe the strengths and weaknesses of the "hei-tagging" method.

We followed this advice and have included a paragraph in the discussion to clearly describe the strengths and weaknesses of “hei-tagging”.

Reviewer #1 (Recommendations for the authors):I have a few concerns regarding the way the authors structure and present their claims:1) One of the main take home messages (based on the Abstract) is that the hei-tag boosts the activity of a "wide variety" of genome editing tools. Even though the BE data are promising, the manuscript could use more examples (e.g. prime editors, that have been used in zebrafish – Petri et al., 2021). Can the hei-tag make editing possible in cases where it is undetectable with the state of the art tools?

We toned down this statement in the abstract. In our new Figure 2 we now show multiplexed editing by using sgRNAs of different efficiencies. In all cases, the heiCas9 performed better (more overall editing, fewer different alleles) compared to the reference zCas9 (Jao et al., 2013).

2) Throughout the manuscript, the authors use only a handful of guides. Adding more examples, especially using guides that show a range of efficiencies with the non-hei-tag Cas9 would provide a better description of the strengths and limitations of the hei-tag.

We followed the referee’s suggestion and have chosen additional sgRNAs of which one targets an intron (*rx3*) and three sgRNAs with different editing activity targeting exonic sequences (*oca2*, *rx2* and *cryaa*). The respective Illumina sequencing analysis is now included as new main Figure 2 and Supplementary file 3.

3) What is the rationale of using the Cas9 from Hwang et al., and not the zebrafish codon optimised Cas9 from Jao et al?

We thank the referee for the suggestion and now use the zCas9 from Jao et al., as reference in our *oca2* analysis pipeline (Figure 1d, g) for which we performed an entirely new set of experiments (transcription, injection and analysis) in medaka and zebrafish. We also use the zCas9 from Jao et al., 2013 as reference in the multiplexing approach in new Figure 2.

4) Why do the authors use the GeneArt nuclease as a "state of the art" for the cell culture? What other tags are included in this nuclease? We can see that the worse score in Fig1d comes from JDS246. Not having the hei-tag is only one difference between JDS246 and heiCas9. One other is that heiCas9 lacks the FLAG (Myc-Cas performs already pretty well, however zebrafish data are not shown). How does GeneArt Cas9 compare in that respect? Such information should be presented to readers.

For cell culture assays, there is only a limited number of commercially available Cas9 nuclease mRNAs to choose from. The exact sequence composition of tags/NLSs of the GeneArt Cas9 mRNA is not further specified by the manufacturer.

5) In Figure 1, what do the single guide results look like? Boosting deletion outcomes using two guides is not the same as boosting activity of single guide editing. Is the activity of both guides improved? Are they just being brought at the same level? Did the authors use sequencing to analyse what happened at the DNA level?

We now analyze the edited alleles with Illumina sequencing (new Figure 2). Since in this approach we followed other suggestions by the referee, we multiplexed different sgRNAs. Therefore, we decided to pick only one sgRNA for *oca2* (T2). We can show that targeting efficiency using heiCas9 clearly improves over the zCas9 from Jao et al., 2013 not only in *oca2* but all targeted loci.

6) The hei-tag could be affecting editing outcomes, rather than ON target activity. In Figure S1, it is clear that the frequency of the in frame -15 allele is reduced, while overall efficiency in not dramatically affected. Changing the editing profile by using different nucleases (variants) is not the same as boosting efficiency in KO. The in-frame indels could be favored in other sites, thus reducing KO-score (if out of frame are regarded as desired outcome).

We agree with the referee that the increased out-of frame repair seen with the heiCas9 may be locus specific. Nevertheless, the induction of indels using the heiCas9 was dramatically higher compared to the other nucleases employed.

7) Generally, it is advised to use NGS data for strong claims about editing efficiency.

We thank the referee and include Illumina sequencing data now in new Figure 2 and Supplementary file 3.

8) If ON-target activity is boosted, increased OFF-target activity is of concern, especially in cell culture when editing accuracy and precision are key. Since the main focus of the authors is not mammalian cell culture or therapeutic applications, they should at least be clear about this limitation.

We discuss this important point now in the main text where we clearly state this potential limitation.

Reviewer #2 (Recommendations for the authors):Several reports in zebrafish have shown that Cas9/gRNA complexes injections can reach gene editing up to 100% in F0 embryos including when targeting multiple loci at the same time (See for instance Wu et al., 2018 and Kroll et al., 2021). In light of these impressive results the utility of the present method for the fish community is overstated at best without mentioning these points anywhere in the discussion.The authors should attempt to compare the injection of heiCAS9 protein as this is the current preferred and most efficient method in the fish community.

We want to kindly point out that especially in the medaka community, protein injection of effectors of most kinds is not the common protocol. In case of targeted genome editing, also Cas9 and BaseEditors are usually provided as mRNA.

Following the referees suggestion, we have now compared the most recent variant of the IDT Cas9 protein (Alt-R S.p.Cas9 V3) in similar concentrations as used by Wu et al., 2018, Dev Cell (5µM) and Kroll et al., 2021, *eLife* (24µM) and equimolar to the mRNA molecule concentration we use (1.765µM) to heiCas9 mRNA injections (new Figure 1d). This clearly demonstrates that in medaka, mRNA injection with the heiCas9 variant is equally effective compared to protein injections.

The benefits of using mRNA over protein injections are: cost-efficiency, flexibility in generating variants, easier handling of the injection mix (24µM and 5µM protein concentrations are very viscous, often leading to needle clogging and hence affecting injection throughput and success).

In the present manuscript is not clear what is the purpose of the Myc tag in the construct. This is not used at all to visualize protein localization or purification. Do the authors believe that the addition of Myc increases Cas9 activity? If not giving that this is a method paper they should demonstrate the utility in this specific contest.

The Myc-tag was initially included to facilitate straight forward protein purification. Strikingly, the injection of mRNA of this construct already dramatically increased knock-out efficiencies eliminating the need for protein purification.

We performed a permutation screen of NLSs, Myc-tag and linkers and addressed the impact on knock-out efficiency. This resulted in the identification of the hei-tag (myc, flexilinker, oNLS) described here. To provide this information to the community, we now include this screen in Figure 1—figure supplement 1.

The zebrafish experiment lacks the comparison of the heicas9 with the most efficient myc-Cas9 leading to an overestimation to the improved efficiency of the construct.

By suggestion of referee #1 we now have changed the reference Cas9 to “the zebrafish codon optimised Cas9 from Jao et al., 2013” in the oca2 knock-out screen and have adapted the medaka injections as well. Interestingly in zebrafish, zCas9 was not significantly better than the earliest Cas9 version (“JDS246”) – see updated Figure 1. In medaka however, the zCas9 performed well but clearly less efficient than the heiCas9.

Finally methodological papers based on a single locus are difficult to appreciate as they may be influenced by "the gambler's luck" (i.e. the chosen locus cold be a fortunate pick). These results should be extended to other genes as it's standard in the field (see the two publications above).

We thank the referee for the point raised here and now included 3 more sgRNAs with different activity and position (intronic/exonic). In all cases, heiCas9 not only performed more efficiently compared to the zCas9 from Jao et al., but also generated fewer different alleles over all modified alleles detected in the Illumina sequencing (see new Figure 2 and Supplementary file 3).

Finally the base editor experiment are equally strongly biased. The comparison with BE4-Gam is not representative of the current state of the art were several reports using ancBE4Max (See for instance Carrington et al., 2020, Zao et al., 2020 and Rosello et al., 2021) show highly improved results. In lights of these papers the statement "Notably, in heiBE4-Gam injections, for each of the three cytosines in the base editing window, the C-to-T transition rate was higher than 60%, a level never observed in BE4-Gam injected embryos" is not true as similar or higher level of C to T conversion have been reported. Again this a comparison with ancBE4Max should be extended to multiple loci.

The referee is right and we changed the indicated sentence. We feel that we need to clarify our rationale here (as we did in the amended main text): it was not our intention to generate “the best editor”. The activity of most recent editors clearly plateaus at the standard concentrations. Our intention rather was to show that a little tag can boost activity, which is easier detected in the exponential phase. We therefore used a version of the BaseEditor that has not reached its activity plateau to address this point. We now clearly state this in the revised version of the manuscript.

Reviewer #3 (Recommendations for the authors):1. In the abstract, the authors introduced factors that leave room for the improvement of gene editing efficiency in CRISPR/Cas9 tools: (1) nuclear localization signal -citing Cong et al., 2013; and (2) protein tags for "immediate detection or straight-forward purification" and linkers to "avoid steric hinderance impacting on activity" -citing Zhang et al., 2014. However, such conclusions were never made in either of the originally cited papers. Cong et al., did not compare the editing activity with and without a NLS signal. On the contrary, there are partial evidence indicating that Cas9 protein may not require an NLS to assist import into the nucleus (Hu et al., G3, 2018). In Zhang et al., it was suspected that the addition of a flag or myc tag changed the charge distribution of the Cas9 protein, thus increasing its specificity and efficacy. No statement was made about the purification or the linker. It is OK to introduce the relevant background, but I find it problematic to misinterpret the cited literature to show conclusions they did not make.

We thank the referee for the comment and have carefully revised the manuscript to avoid the problems indicated.

2. The authors used modified version of NLS in the hei-tag construct to facilitate early nuclear targeting, while a straight-forward way to make nuclear-targeted Cas9 available in the cell is to directly inject the (nuclear-targeting) Cas9 protein. The authors should either provide this control experiment, or clarify why they only chose to build RNA-based systems.

We now have included a comparison to protein/sgRNA complex injections in Figure 1. Following the referees suggestion, we have now compared the most recent variant of the IDT Cas9 protein (Alt-R S.p.Cas9 V3) in similar concentrations as used by Wu et al., 2018, Dev Cell (5µM) and Kroll et al., 2021, *eLife* (24µM) and equimolar to the mRNA molecule concentration we use (1.765µM) to heiCas9 mRNA injections (new figure 1d). This clearly demonstrates that in medaka, mRNA injection with the heiCas9 variant is equally effective compared to protein injections.

The benefits of using mRNA over protein injections are: cost-efficiency, flexibility in generating variants, easier handling of the injection mix (24µM and 5µM protein concentrations are very viscous, often leading to needle clogging and hence affecting injection throughput).

3. It is widely known that the gRNA design is a critical factor affecting the gene editing efficiency. While hei-tag shows an increased bi-allelic editing efficiency than the control constructs, it is not clear whether this boost is universal with different targeted genes and different sgRNA designs.

We followed the referee’s suggestion and have chosen additional sgRNAs of which one targets an intron (rx3) and three sgRNAs with different editing activity targeting exonic sequences (oca2, rx2 and cryaa). The respective Illumina sequencing analysis is now included as new main figure 2 (and Supplementary file 3). In all cases, heiCas9 outperformed the reference.

4. JDS246-Cas9 was chosen as the baseline construct to evaluate the boost of editing efficiency. Given this was a construct originally made for mammalian cells, it is not clear whether it represents the state-of-art editing techniques, especially in zebrafish. While it is unrealistic to test all the available tools, other systems have been reported with high bi-allelic editing efficiency specifically in zebrafish should be introduced as a control (e.g. Jao et al., 2013, PNAS -disclaimer: the reviewer was not a maker of this tool).

We now have included the comparison to “the zebrafish codon optimised Cas9 from Jao et al.” and have adapted the medaka injections as well (Figure 1). Interestingly, it was not significantly better than the earliest mammalian optimized Cas9 version (“JDS246”) – see updated Figure 1. In medaka however, the zCas9 performed very well but clearly less efficient than the heiCas9. The Cas9 we use in our constructs is mammalian codon optimized, and stellar also in zebrafish.

5. In the base-editing experiments, the injected fish showed various level of eye-pigmentation colors, in contrast to the knockout experiment where cells devoid of pigment appear in patches on the eye. The authors should provide an explanation of why this is the case and justify why the pigmentation level has to be quantified differently in Figure 3 than Figure 1.

In both experiments, the loss of pigmentation phenotype is identical – loss of pigmentation appears in patches. We repeated the BaseEditor experiments and quantified according to the procedure demonstrated in Figure 1.